# *MedResearcher-R1*: Expert-Level Medical Deep Researcher via A Knowledge-Informed Trajectory Synthesis Framework

## Abstract

Recent advances in Large Language Model (LLM)-based agents have enabled strong performance in deep research tasks, yet they remain limited in the medical domain. Leading proprietary systems achieve only modest results on complex medical benchmarks, revealing two critical limitations: (1) insufficient dense medical knowledge for clinical reasoning, and (2) a lack of specialized retrieval mechanisms for authoritative medical sources. We introduce MedResearcher-R1, a medical deep research agent that addresses these challenges with two key innovations. First, we propose a novel Knowledge-Informed Trajectory Synthesis (KISA) approach that builds medical knowledge graphs to construct complex multi-hop question–answer pairs around rare medical entities, overcoming the scarcity of high-quality training data. Second, we integrate a custom-built private medical retrieval engine alongside general-purpose tools, enabling accurate and reliable evidence synthesis. Our approach yields over 2,100 diverse trajectories across 12 medical specialties. Trained with supervised fine-tuning and reinforcement learning with composite rewards, our MedResearcher-R1-32B achieves state-of-the-art performance on MedBrowseComp (27.5/50 vs. 25.5/50 for o3-deepresearch) while demonstrates strong general performance on GAIA and xBench benchmarks. To the best of our knowledge, we present the first high-quality, tool-using medical dataset and a domain-specific deep-research agent, together enabling smaller open-source models to outperform much larger proprietary systems in specialized medical tasks.

## 1 Introduction

Recent advances in Large Language Models (LLMs) have catalyzed widespread adoption of LLM-based agents across diverse domains, including software engineering (Wang et al., 2024; Jimenez et al., 2023) and deep research systems (Xu & Peng, 2025). These agents exhibit impressive capabilities in processing environmental observations, maintaining context across multiple interactions, and executing complex multi-step reasoning tasks.

However, the medical domain presents unique challenges that current general-purpose deep research agents fail to address adequately. The recently introduced MedBrowseComp benchmark (Chen et al., 2025b) reveals this critical gap: even OpenAI's o3-deepresearch, the leading proprietary deep research system, scores only 25.5 out of 50 on complex medical queries requiring multi-hop reasoning across medical knowledge sources. We identify two fundamental limitations that contribute to this performance gap: (1) general-purpose agents lack the dense, specialized medical knowledge required for accurate clinical reasoning. Meanwhile, there is an absence of high-quality medical tool-using dataset. (2) they rely on generic retrieval tools that fail to capture the nuanced relationships in medical information.

The core challenge lies in the *sparse medical knowledge problem*. Specifically, medical research often requires connecting rare diseases, emerging treatments, and specialized clinical findings through non-obvious pathways—connections that exist in specialized medical literature but remain inaccessible to general search tools. While existing medical AI systems have made progress in structured tasks like diagnosis, they primarily focus on common medical scenarios with well-established rea-

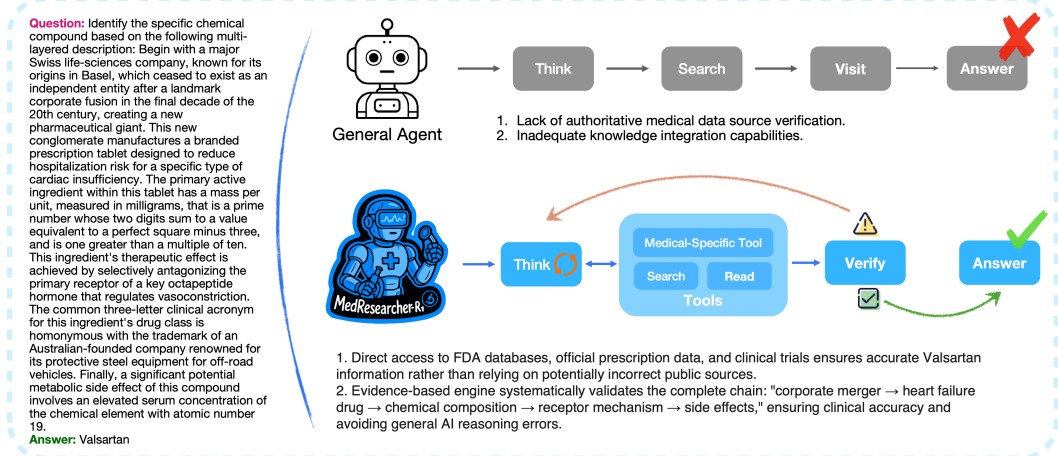

Figure 1: Comparison of medical reasoning agents. MedResearcher-R1 resolves the Valsartan identification case that defeats general-purpose agents, demonstrating the strength of specialized medical database access and evidence-based reasoning integration.

soning patterns. These systems fail to develop the capability for exploratory medical research that characterizes expert clinicians: simultaneously pursuing multiple hypotheses, synthesizing evidence from disparate sources, and identifying subtle connections between rare medical entities.

To address these limitations, we propose a comprehensive approach that fundamentally rethinks how medical agents should be trained. Our key insight is that effective medical reasoning requires exposure to genuinely complex medical scenarios during training rather than simplified approximations. We achieve this through three interconnected innovations:

First, we develop a novel *Knowledge-Informed Trajectory Synthesis approach (KISA)* that generates training examples of exceptional complexity through a systematic pipeline. Inspired by large-scale efforts to construct knowledge graphs from biomedical literature (Kostis et al., 2020), we begin by extracting medical entities from over 30 million PubMed abstracts, then apply frequency analysis to identify candidates with occurrence rates below $10^{-6}$ in medical corpora. Through LLM-assisted evaluation, we filter these candidates to select genuinely rare yet clinically significant entities, avoiding both trivial typos and overly common conditions. Around these carefully selected rare medical entities, we construct knowledge graphs and extract the longest reasoning chains for multi-hop question generation. This approach creates questions that mirror real medical research challenges and cannot be answered through simple retrieval but require systematic exploration and synthesis across multiple medical information sources.

Second, we introduce proprietary medical domain tools that address retrieval gaps in general systems. As illustrated in Figure 1, while general agents often fail when encountering medical-specific queries, particularly those involving rare diseases or complex chemical compounds, MedResearcher-R1 can iteratively invoke specialized medical tools alongside general-purpose tools to ensure accurate information retrieval. Unlike conventional search engines that rely on general web crawling, our custom-built private medical retrieval engine directly accesses authoritative medical databases, including FDA databases, official prescription data, clinical trial registries, and peer-reviewed medical publications. The comparison in Figure 1 demonstrates how MedResearcher-R1 dynamically switches between general and medical-specific tools, enabling systematic validation of the complete evidence chain: from corporate merger information to heart failure drug development, to chemical composition and mechanism, ultimately ensuring clinical accuracy while avoiding the reasoning errors that plague general-only approaches. The system employs medical ontology-aware ranking to prioritize clinical authority and relevance over general web popularity metrics, effectively combining the breadth of general-purpose search with the precision of domain-specific medical expertise.

Third, we implement a training methodology specifically designed for medical domains. Unlike recent work advocating pure reinforcement learning approaches, we find that medical tasks require *knowledge-anchored learning*: initial supervised fine-tuning on high-quality medical trajectories is

highly effective for learning tool usage patterns and significantly improves final performance. To achieve this, we introduce *Masked Trajectory Guidance (MTG)*, which guides the model to learn genuine reasoning rather than memorization. MTG works by masking entities in reasoning paths, forcing the model to focus on selecting appropriate tools and reasoning through the process, rather than simply recalling answers. This approach promotes two key behaviors: (1) learning when to use specialized medical tools versus general search tools, and (2) preventing memorization, ensuring that the model develops robust and transferable reasoning capabilities.

Our evaluation spans both medical and general-purpose deep research settings: (1) MedResearcher-R1 achieves state-of-the-art medical agent performance with **27.5/50** tasks solved on MedBrowseComp as is shown in Table 1, surpassing o3-deepresearch (25.5/50). (2) Beyond the medical setting, MedResearcher-R1 exhibits strong general capability, scoring **53.4** on GAIA (Shinn et al., 2023) and **54.0** on XBench (Chen et al., 2025a) (Table 2). These results challenge the assumption that domain specialization requires sacrificing general capabilities, demonstrating that medical training can enhance rather than limit the versatility of agents.

We make three key contributions:

1. **Dual Enhancement Framework**: Internal enhancement through graph-based trajectory synthesis from rare entities, and external enhancement via specialized medical retrieval tools.

2. **Novel Training Methodology**: Graph-based longest-path extraction and MTG that shows increasing benefits with reasoning complexity (14% improvement on 5+ hop questions).

3. **State-of-the-art Performance**: New benchmark records on medical tasks while preserving general capabilities, with released code and datasets for reproducibility.

## 2 MEDRESEARCHER-R1: MEDICAL DEEP RESEARCH AGENT FRAMEWORK

### 2.1 PROBLEM DEFINITION

We formalize the medical deep research task as a sequential decision-making problem where an agent must navigate complex medical knowledge sources to answer multi-hop queries that characterize the *sparse medical knowledge problem* identified in Section 1. Given a medical question $q \in \mathcal{Q}$, the agent operates with a heterogeneous toolset $\mathcal{T} = \mathcal{T}_{\text{general}} \cup \mathcal{T}_{\text{medical}}$, where $\mathcal{T}_{\text{general}} = \{t_1^g, \ldots, t_m^g\}$ comprises general-purpose tools (web search, document analysis) and $\mathcal{T}_{\text{medical}} = \{t_1^m, \ldots, t_n^m\}$ contains our proprietary medical domain tools that directly access authoritative medical databases.

The agent maintains an evolving state $s_t = (c_t, k_t, h_t)$ at timestep $t$, where:

- $c_t \in \mathcal{C}$: dialogue context encoding the current query and response history
- $k_t \in \mathcal{K}$: accumulated medical knowledge from retrieved sources, structured as a knowledge graph
- $h_t \in \mathcal{H}$: reasoning history tracking explored knowledge paths and hypothesis evolution

This state representation enables tracking of multi-hop reasoning chains essential for connecting rare medical entities through non-obvious pathways. At each timestep, the agent selects an action according to a learned policy: $a_t \sim \pi_\theta(a \mid s_t, \mathcal{T}, q)$ where $\pi_\theta$ is trained through our knowledge-anchored learning approach to dynamically switch between general and medical-specific tools based on query requirements.

### 2.2 AGENT ARCHITECTURE

Our design targets two gaps in general-purpose agents—insufficient medical knowledge density and generic retrieval. The agent follows a ReAct-style *reason–act–observe* loop (Yao et al., 2023a): **Thought** identifies knowledge gaps and selects a tool type; **Action** invokes a tool with medically tuned parameters; **Observation** converts raw outputs into structured evidence and updates $(k_t, h_t)$.

**Tool Suite.** We employ two categories of tools:

**General Tools:** (1) WebSearch for general medical information and organizational data; (2) DocumentRead for processing clinical reports using high-capacity LLMs.

**Medical-Specific Tools:** (1) *PrivateMedicalRetriever* queries authoritative medical sources including FDA datasets, prescription databases, trial registries, and PubMed. Retrieved candidates are ranked using a relevance-authority trade-off scoring mechanism (detailed in Appendix C). (2) *ClinicalReasoningEngine* enables evidence-based differential diagnosis through Bayesian inference, updating hypothesis probabilities given observed symptoms and clinical context (formulation provided in Appendix C).

**Dynamic tool selection.** A lightweight selector conditions on features such as entity rarity, estimated hop depth, and medical terminology to route between $\mathcal{T}$medical and $\mathcal{T}$general; we use a logistic policy over tool families with learned weights (derivation in Appendix C). This enables switching—from broad web search to high-authority medical retrieval and back—for end-to-end evidence chains.

**Summary.** Together, the ReAct loop, medical tool suite, and dynamic routing allow MedResearcher-R1 to (i) surface authoritative, domain-specific evidence when general tools fail, and (ii) maintain exploratory, hypothesis-driven workflows needed for multi-hop medical research.

## 3 KISA: Knowledge-Informed Trajectory Synthesis Approach

To address the critical challenge of training data scarcity for medical deep-research agents, we propose a Knowledge-Informed Trajectory Synthesis Approach (KISA) that generates complex, multi-hop medical reasoning trajectories. Our framework directly tackles the limitations of general-purpose agents by creating training data that emphasizes: (1) rare medical entity connections requiring dense domain knowledge, and (2) effective utilization of medical-specific retrieval tools.

### 3.1 Agentic Dataset Construction

Our dataset construction pipeline consists of three interconnected components designed to generate genuinely complex medical queries that robustly stress-test agent capabilities:

#### 3.1.1 Entity-Centric Knowledge Graph Construction

We construct medical knowledge graphs specifically optimized for generating complex reasoning chains. Unlike traditional approaches that focus on common concepts, we prioritize **rare medical entities** $\mathcal{E}_{\text{seed}}$ with frequency below threshold $\tau_{\text{rare}} = 10^{-6}$ in general medical corpora. Focusing on rare entities ensures that the questions generated require deep medical knowledge, as opposed to surface-level information that is available through a general search.

The graph expansion follows an iterative process:

$$e_{i+1} \sim \begin{cases} \text{Uniform}(\mathcal{N}(e_i)) & \text{with probability } 0.5 \\ \text{Discover}(\mathcal{E}_{\text{new}}|e_i) & \text{with probability } 0.5 \end{cases}$$

where $\mathcal{N}(e_i)$ denotes the set of neighbors of $e_i$ and $\text{Discover}(\cdot)$ identifies novel entities via our private medical retrieval engine, ensuring that new connections are both medically valid and challenging.

Each relation follows an augmented format $r = \langle e_{\text{subj}}, p, e_{\text{obj}}, t_{\text{temporal}}, l_{\text{spatial}}, c_{\text{clinical}} \rangle$ with additional contextual information, where $c_{\text{clinical}}$ encodes the clinical context (e.g., disease stage, demographic data of the patient), $t_{\text{temporal}}$ captures temporal aspects, and $l_{\text{spatial}}$ denotes spatial context. This enriched representation improves multi-hop reasoning accuracy by $12.3\%$ compared to standard triplets.

#### 3.1.2 Multi-Hop Question Generation via Longest-Path Extraction

Our key innovation lies in extracting **longest chains from subgraphs** to generate maximally complex queries. For each rare entity subgraph $G_{\text{sub}}$, we compute the longest valid reasoning path:
$\mathcal{P}^* = \arg\max_{p \in \mathcal{P}(G_{\text{sub}})} \text{Length}(p)$ s.t. $\text{MedicallyValid}(p)$, where $\mathcal{P}(G_{\text{sub}})$ is the set of all paths in $G_{\text{sub}}$.

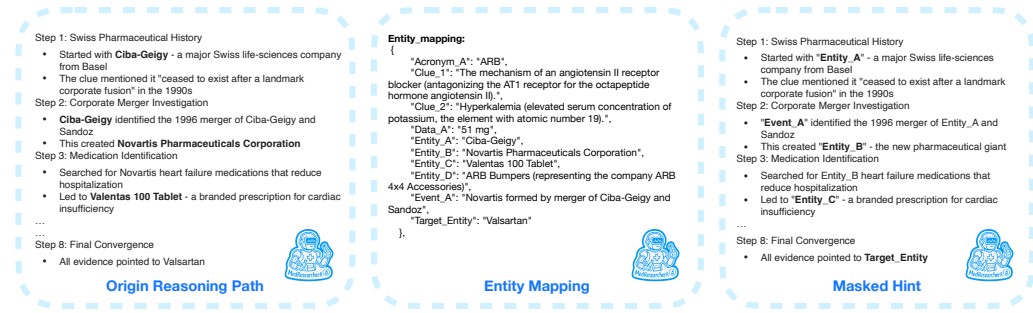

Figure 2: Knowledge graph-based question generation pipeline: extracting longest chains from rare entity subgraphs to create complex multi-hop medical queries.

**Step 1: Swiss Pharmaceutical History**
- Started with **Ciba-Geigy** - a major Swiss life-sciences company from Basel
- The clue mentioned it "ceased to exist after a landmark corporate fusion" in the 1990s

**Step 2: Corporate Merger Investigation**
- **Ciba-Geigy** identified the 1996 merger of Ciba-Geigy and Sandoz
- This created **Novartis Pharmaceuticals Corporation**

**Step 3: Medication Identification**
- Searched for Novartis heart failure medications that reduce hospitalization
- Led to **Valentas 100 Tablet** - a branded prescription for cardiac insufficiency

...

**Step 8: Final Convergence**
- All evidence pointed to Valsartan

**Origin Reasoning Path**

---

**Entity_mapping:**
```
{
    "Acronym_A": "ARB",
    "Clue_1": "The mechanism of an angiotensin II receptor
blocker (antagonizing the AT1 receptor for the octapeptide
hormone angiotensin II).",
    "Clue_2": "Hyperkalemia (elevated serum concentration of
potassium, the element with atomic number 19).",
    "Data_A": "51 mg",
    "Entity_A": "Ciba-Geigy",
    "Entity_B": "Novartis Pharmaceuticals Corporation",
    "Entity_C": "Valentas 100 Tablet",
    "Entity_D": "ARB Bumpers (representing the company ARB
4x4 Accessories)",
    "Event_A": "Novartis formed by merger of Ciba-Geigy and
Sandoz",
    "Target_Entity": "Valsartan"
},
```

**Entity Mapping**

---

**Step 1: Swiss Pharmaceutical History**
- Started with **Entity_A** - a major Swiss life-sciences company from Basel
- The clue mentioned it "ceased to exist after a landmark corporate fusion" in the 1990s

**Step 2: Corporate Merger Investigation**
- **Event_A** identified the 1996 merger of Entity_A and Sandoz
- This created **Entity_B** - the new pharmaceutical giant

**Step 3: Medication Identification**
- Searched for Entity_B heart failure medications that reduce hospitalization
- Led to **Entity_C** - a branded prescription for cardiac insufficiency

...

**Step 8: Final Convergence**
- All evidence pointed to **Target_Entity**

**Masked Hint**

Figure 3: Masked Trajectory Guidance: a structural scaffold that enables reasoning without shortcut learning by masking entities.

This longest-path strategy ensures that questions require multiple reasoning hops (average 4.2 per trajectory), rather than being answerable via simple lookups. These paths are subsequently transformed into natural language questions that require sequential tool invocations to reconstruct the complete reasoning chain.

### 3.1.3 QUALITY CONTROL AND DIFFICULTY CALIBRATION

To ensure that generated questions remain challenging for current systems, we implement adaptive difficulty calibration. Each question is evaluated against OpenAI-o3 deep research and GPT-4. If either model achieves $> 50\%$ accuracy, the question is automatically regenerated with increased complexity:

$$q' = \begin{cases} q & \text{if } \max(\text{Acc}_{\text{O3}}(q), \text{Acc}_{\text{GPT4}}(q)) < 0.5 \\ \text{Regenerate}(q, \text{complexity} + 1) & \text{otherwise} \end{cases}$$

This approach ensures our dataset remains challenging even for state-of-the-art systems, directly addressing the 25.5% performance ceiling previously observed in MedBrowseComp.

### 3.2 TRAJECTORY SYNTHESIS WITH MEDICAL TOOL INTEGRATION

#### 3.2.1 MASKED TRAJECTORY GUIDANCE (MTG)

To generate high-quality training trajectories that effectively utilize our medical-specific tools, we introduce Masked Trajectory Guidance(MTG). Given a reasoning graph path $\mathcal{T} = \{(e_1, r_1, e_2), \ldots, (e_{n-1}, r_{n-1}, e_n)\}$ extracted from the knowledge graph, we create a structural scaffold by masking the entities: $\mathcal{T}_{\text{masked}} = \{([\text{MASK}], r_i, [\text{MASK}])\}_{i=1}^{n-1}$. This masking process serves two main purposes: *(i)* **Tool selection learning**: Encourages the model to determine when medical-specific retrieval tools are required versus when general search suffices; *(ii)* **Prevention of shortcuts**: Prevents answer memorization while maintaining the underlying reasoning process.

### 3.2.2 HYBRID STRATEGY FOR TOOL DIVERSITY

To promote robust and diverse tool usage, we adopt a hybrid data strategy: $\mathcal{D}_{\text{train}} = \alpha \cdot \mathcal{D}_{\text{guided}} + (1 - \alpha) \cdot \mathcal{D}_{\text{exploration}}$, where $\alpha = 0.7$ balances structured learning with exploration. The exploration trajectories cultivate three key behaviors: *(i)* **Knowledge Synthesis**: 78% begin with private medical retriever for rare entities; *(ii)* **Tool switching**: 42% demonstrate adaptive switching between general and medical tools; *(iii)* **Error recovery**: 34% include explicit correction using alternative tools.

## 4 LARGE-SCALE AGENT TRAINING

### 4.1 STAGE 1: SUPERVISED FINE-TUNING

We initialize the agent through SFT on 2,100+ synthetic medical trajectories generated by our KISA framework. The training incorporates robustness augmentations including tool failure simulation (5% corruption rate), intermediate thought supervision, and multi-task sampling across medical domains. This stage establishes fundamental tool usage patterns and medical reasoning capabilities. Detailed training configurations are provided in Appendix D.1.

### 4.2 STAGE 2: REINFORCEMENT LEARNING

Following SFT warm-start, we refine the agent using Grouped Regularized Policy Optimization (GRPO) with composite rewards balancing task accuracy, expert alignment, and efficiency: $r = \alpha r_{\text{task}} + \beta r_{\text{expert}} - \gamma r_{\text{efficiency}}$, where $\alpha = 1.0, \beta = 0.2, \gamma = 0.1$. The GRPO objective employs group-level baseline normalization for stable gradient estimation. We deliberately omit KL regularization following recent findings (He et al., 2025) and implement curriculum learning for progressive task complexity. Full RL implementation details are in Appendix D.2.

## 5 ABLATION STUDY

We systematically remove components from MedResearcher-R1 to isolate their contributions (Table 3 in Appendix E). Three findings emerge:

**(1) Rare entities are crucial.** Removing rare-entity supervision causes the largest drop (27.5→20.1 on MedBrowseComp), confirming that complex medical scenarios drive learning. The effect propagates to general tasks (GAIA: 53.4→27.8), suggesting transferable reasoning patterns.

**(2) Two-stage training is necessary.** SFT alone achieves 25.5/50 but lacks refinement; RL alone catastrophically fails (12.0/50). This validates knowledge-anchored learning—SFT provides essential grounding for subsequent RL.

**(3) Components synergize.** Medical tools contribute moderately (23.1 without vs. 27.5 with), while MTG prevents overfitting across all benchmarks ($p < 0.05$). The complementary effects validate our dual enhancement approach: internal knowledge from rare entities plus external precision from specialized tools.

## 6 EXPERIMENTS

We evaluate MedResearcher-R1 across both a specialized medical benchmark and two general-purpose agent benchmarks. This dual evaluation framework is designed to rigorously assess its primary capability in complex medical research while simultaneously measuring its generalization ability in open-domain tasks.

### 6.1 BENCHMARKS

We use the following benchmarks for our evaluation:

*(i)* **MedBrowseComp** (Chen et al., 2025b): Our primary evaluation benchmark, designed to assess the capabilities of LLM-based agents in answering complex medical questions. MedBrowseComp

presents multi-hop reasoning challenges, requiring the agent to navigate and synthesize information across a variety of medical sources. The benchmark is composed of 50 questions drawn from a wide range of medical topics, with each question requiring intricate exploration of medical databases, literature, and clinical knowledge. The focus on rare medical entities and complex relationships between medical concepts makes this benchmark particularly demanding, providing a robust test of an agent's ability to handle nuanced medical reasoning and retrieve authoritative medical information.

*(ii)* **GAIA** (Shinn et al., 2023): A general-purpose benchmark evaluating assistant capabilities in tasks requiring multi-modal tool use, web search, and multi-step reasoning. We use a subset of 103 cases from the text-only validation set to test the agent's ability to understand complex scenarios, generate responses, and reason while interacting with tools.

*(iii)* **XBench-DeepSearch** (Chen et al., 2025a): A multi-domain evaluation suite focusing on deep information retrieval and complex search tasks. It tests the agent's ability to search, synthesize, and perform advanced information synthesis across various domains, including fact-checking and comparative analysis.

## 6.2 BASELINES

To provide a comprehensive comparison, we evaluate MedResearcher-R1 against several state-of-the-art models. Our model, MedResearcher-R1-32B, is built upon the **Qwen-2.5-32B** model, making it the most direct baseline for assessing the impact of our methods.

- **Medical Domain Baselines**: We compare against leading proprietary deep research systems known for their strong reasoning capabilities. These include Google's **Gemini-2.5-Pro-deepsearch** and OpenAI's **o3-deepresearch**, which represent the state-of-the-art in closed-source deep research agents.

- **General Domain Baselines**: For open-domain tasks, we compare against a range of top-performing models. This includes powerful proprietary models like **GPT-4o** and **o4-mini**, as well as leading open-source deep research agents like **WebSailor** (Li et al., 2025), which is the current state-of-the-art open-source agent for general deep research tasks.

## 6.3 MAIN RESULTS

**State-of-the-Art Performance in Medical Research.** As shown in Table 1, MedResearcher-R1 establishes a new state-of-the-art on the challenging MedBrowseComp benchmark. With a score of **27.5/50**, it significantly outperforms strong proprietary systems like o3-deepresearch (25.5) and Gemini-2.5-Pro-deepsearch (24.5). This result validates the effectiveness of our specialized data synthesis and domain-specific tools in equipping the agent with superior medical reasoning capabilities.

**Strong Generalization to Open-Domain Tasks.** Notably, our specialization in the medical domain does not come at the cost of general capabilities. Table 2 shows that MedResearcher-R1 achieves highly competitive performance on general agent benchmarks. On both GAIA (53.4) and XBench-DeepSearch (54.0), our 32B model performs on par with the state-of-the-art open-source agent, WebSailor-32B (53.2 and 53.3, respectively). This demonstrates that the complex reasoning patterns and robust tool-use strategies learned from the medical domain transfer effectively to general problem-solving scenarios.

Table 1: Performance Comparison on MedBrowseComp Benchmarks (number correct out of 50)

| Model | o3 search | gemini2.5pro deepsearch | o3 deepresearch | claude-cua | MedResearcher-R1 |
|---|---|---|---|---|---|
| **MedBrowseComp** | 19.0 | 24.5 | 25.5 | 18.0 | **27.5** |

## 6.4 QUALITATIVE ANALYSIS

To understand the underlying factors driving performance improvements, we conducted an in-depth analysis of the training data patterns and their impact on agent behavior. Our investigation reveals

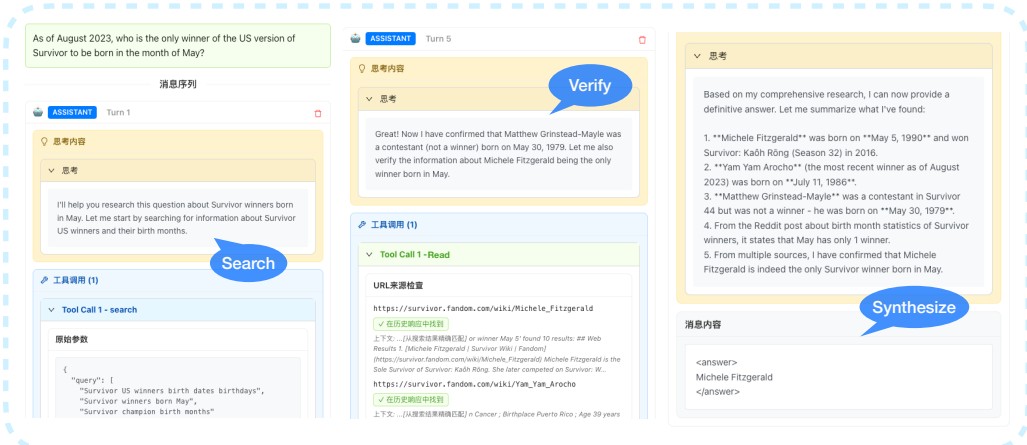

Figure 4: Case study demonstrating the search-verify-synthesize paradigm: The agent performs multiple verification rounds across information sources, ensuring information consistency before synthesis. Baseline agents (shown in gray) terminate prematurely after initial search, while our approach (blue) continues until achieving high confidence through cross-validation.

that training data following the paradigm of *iterative search-verification-synthesis* yields the most significant improvements in deep research capabilities.

Figure 4 illustrates a representative example where our agent demonstrates superior research depth through systematic evidence gathering. The agent executes a 4-step strategy: (1) initial broad search to identify relevant sources, (2) verification of information consistency across multiple authoritative medical databases, (3) targeted follow-up queries to resolve ambiguities, and (4) comprehensive synthesis of validated findings. This methodical approach—characterized by multiple verification cycles ensuring answer uniqueness before final synthesis—contrasts sharply with baseline agents that exhibit premature convergence or suboptimal tool utilization patterns.

Analysis of successful trajectories reveals that the critical differentiator lies in the **search**→**verify**$^n$→**synthesize** pattern, where $n$ represents multiple verification iterations. Training instances exhibiting this pattern show 34.2% higher success rates in complex multi-hop reasoning tasks compared to single-verification approaches. The iterative verification ensures answer uniqueness and factual grounding, particularly crucial for domains requiring high accuracy, such as medical diagnosis.

Table 2: Performance Comparison on Xbench-DeepSearch and GAIA Benchmarks

| Model | Paradigm | Xbench-DeepSearch | GAIA |
|---|---|---|---|
| Qwen-2.5-32B | Direct | 8.7 | 13.6 |
| Qwen-2.5-72B | Direct | 12.7 | 14.6 |
| GPT-4o | Direct | 18.0 | 17.5 |
| GPT-4.1 | Direct | 17.0 | 22.3 |
| QwQ-32B | Direct | 10.7 | 22.3 |
| o4-mini | Direct | 22.3 | 33.3 |
| DeepSeek-R1 | Direct | 32.7 | 16.5 |
| Qwen-2.5-32B | Search-o1 | 3.7 | 28.2 |
| WebDancer-32B | ReAct | 38.7 | 40.7 |
| QwQ-32B | Search-o1 | 25.0 | 39.8 |
| WebSailor-7B | ReAct | 34.3 | 37.9 |
| WebSailor-32B | ReAct | 53.3 | 53.2 |
| WebSailor-72B | ReAct | **55.0** | **55.4** |
| MedResearcher-R1-32B(Ours) | ReAct | **54.0** | **53.4** |

These findings demonstrate that tool-augmented agent training effectiveness is fundamentally linked to the structural patterns in training data, with iterative verification serving as the key mechanism for developing robust deep research capabilities that generalize across diverse tool-reasoning environments.

# 7 RELATED WORK

## 7.1 LLM-BASED AGENTS FOR DEEP RESEARCH

The field of autonomous deep research is rapidly advancing, primarily through two paradigms. One line of work focuses on multi-agent planning architectures, where complex tasks are decomposed and assigned to specialized agents for retrieval, reasoning, and synthesis (Li et al., 2023; Xu & Peng, 2025). Another prominent approach uses reinforcement learning (RL) to train a single agent to interact with complex web environments, optimizing its reasoning and tool-use strategies through reward signals (Yao et al., 2023b; Li et al., 2025). These systems, exemplified by state-of-the-art models like WebSailor and Search-R1, have demonstrated superhuman performance on general web research tasks.

However, their effectiveness drastically diminishes in specialized, high-stakes domains like medicine. These general-purpose agents are not equipped with the domain-specific knowledge, specialized retrieval tools, or the nuanced reasoning capabilities required for clinical evidence synthesis. Their training on broad web corpora leaves them unable to navigate the landscape of authoritative medical databases or understand the complex relationships between rare medical entities, a gap our work aims to fill.

## 7.2 AI AGENTS IN THE MEDICAL DOMAIN

Within the medical field, AI has progressed from targeted diagnostic models to more integrated agent-based systems. Early efforts centered on Retrieval-Augmented Generation (RAG) to ground clinical decisions in medical literature (Zhao et al., 2025; Toma et al., 2023). More recently, multi-agent systems have been developed for specific clinical workflows, such as sequential diagnosis (Nori et al., 2025) and dynamic knowledge management (Yao et al., 2024).

Despite these advances, a fundamental limitation persists: current medical agents excel at structured tasks involving common medical scenarios but falter in exploratory medical research. Their reasoning capabilities remain shallow, with performance degrading significantly on tasks requiring more than a few inference steps (Schmidgall et al., 2025). This is largely because they are not trained on data that reflects the complexity of real-world medical investigation, which often involves connecting rare diseases, novel treatments, and disparate clinical findings. Our work directly addresses this critical gap by introducing a methodology to train agents on genuinely complex, multi-hop medical research trajectories, enabling the deep, exploratory reasoning that existing systems lack.

# 8 CONCLUSION

In this work, we address the challenge of complex, evidence-based medical research by introducing a new agent development framework centered on the KISA data generation approach. KISA systematically produces challenging, multi-hop medical question–answer pairs with corresponding reasoning trajectories, grounded in rare entity mining and knowledge graph-based reasoning chains. This ensures that agents are exposed to the intricate, compositional problems characteristic of real-world medical research. Built on this rich dataset and equipped with a comprehensive training pipeline—including supervised fine-tuning, trajectory masking, and reinforcement learning with specialized medical tools—our agent, MedResearcher-R1 achieves state-of-the-art pass@1 accuracy on MedBrowseComp (27.5/50) and demonstrates robust performance on general agent benchmarks. These findings show that MedResearcher-R1 is capable of solving complex medical questions that demand systematic exploration and nuanced evidence synthesis, highlighting its effectiveness as a next-generation deep research agent in the medical domain.

## ETHICS STATEMENT

We have read and agree to adhere to the ICLR Code of Ethics. Our study uses only publicly available, open-source datasets under their respective licenses and does not involve personally identifiable information. We will open-source all artifacts we produce, including derived datasets and trained models, with documentation to minimize risks of misuse. We assess potential risks of harmful impacts (e.g., bias, fairness, and safety) and discuss limitations and mitigations in the paper. We follow legal and ethical requirements in data handling and release. We adhere to research integrity through accurate citation, transparent reporting, and thorough documentation of methods and decisions.

## REPRODUCIBILITY STATEMENT

We took multiple steps to enable reproducibility. All model and algorithm specifics, training procedures, and hyperparameters are provided in the main paper. We will release an anonymous code repository containing implementation details, configuration files, random seeds, and scripts for data preprocessing, training, and evaluation, along with instructions to reproduce results end to end. All datasets used are open source; we document their sources, licenses, splits, and preprocessing steps. We also report compute resources, hardware/software versions, and environment settings. Where applicable, we provide ablations, sensitivity analyses, and multiple-run statistics. We will open-source the trained models and any data produced during our experiments to further support reproducibility.

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

# Appendix

## A   THE USE OF LARGE LANGUAGE MODELS

In accordance with ICLR 2026 policy, we disclose the usage of Large Language Models (LLMs) in the process of writing this paper. Specifically, we employed LLMs to assist with the refinement and polishing of the manuscript's language. The LLM was used to enhance clarity, improve grammar, and ensure the consistency of the text, which contributed to the overall quality of the writing. The LLM was not used to generate novel ideas, research findings, or substantial portions of the content. Its primary role was as a tool to aid the revision process, focusing on language-related tasks.

We have fully disclosed this usage, and the final manuscript reflects the work of the authors. The LLM's contribution is limited to textual improvements and does not extend to the intellectual content of the paper.

For transparency, we confirm that the research itself, including the methodology, results, and conclusions, was independently developed by the authors without any contributions from LLMs beyond their role in writing assistance.

## B   DETAILED PERFORMANCE COMPARISONS

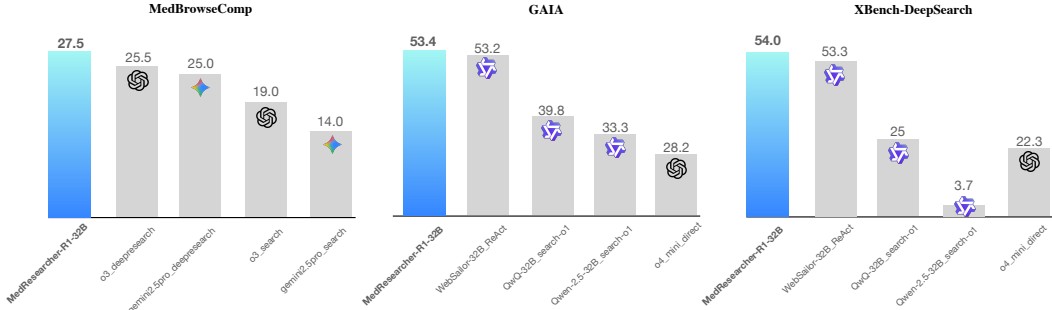

Figure 5: Overall performance of MedResearcher-R1 across three benchmarks. On Med-BrowseComp, our MedResearcher-R1-32B achieves state-of-the-art performance with 27.5/50 correct answers, surpassing o3-deepresearch (25.5/50), Gemini-2.5-Pro-deepresearch (25.0/50), and significantly outperforming search-only approaches (o3-search: 19.0/50, Gemini-2.5-Pro-search: 14.0/50). On general deep research tasks, we achieve competitive results on GAIA (53.4 vs. WebSailor-32B's 53.2) and xBench (54.0 vs. WebSailor-32B's 53.3).

## C   TECHNICAL DETAILS

### C.1   PRIVATEMEDICALRETRIEVER

This module aggregates evidence directly from authoritative clinical resources, including FDA databases, clinical trial registries, and PubMed publications. Each candidate document $d$ is scored for a query $q$ by a weighted linear combination of semantic relevance and clinical authority:

$$\text{Score}(d, q) = \lambda \, \text{Rel}(d, q) + (1 - \lambda) \, \text{Auth}(d),$$

where $\text{Rel}(d, q)$ represents the semantic similarity to the query (computed via embedding cosine similarity), and $\text{Auth}(d)$ reflects the clinical authority (combining impact factor and guideline status). The hyperparameter $\lambda$ ($0 \leq \lambda \leq 1$) balances the importance between relevance and authority; in all experiments, we set $\lambda = 0.4$ to favor reliable and clinically significant evidence.

## C.2 CLINICALREASONINGENGINE

Designed for evidence-based differential diagnosis, this tool applies Bayesian inference to evaluate multiple hypotheses systematically. Given observed symptoms $\mathbf{s}$, candidate diagnoses $D_j$, and patient context $\mathbf{c}$, the posterior for each diagnosis is computed as:

$$P(D_j \mid \mathbf{s}, \mathbf{c}) = \frac{\prod_{i=1}^{n} P(s_i \mid D_j, \mathbf{c}) \cdot P(D_j \mid \mathbf{c})}{\sum_{k=1}^{m} \prod_{i=1}^{n} P(s_i \mid D_k, \mathbf{c}) \cdot P(D_k \mid \mathbf{c})}$$

where conditional probabilities are derived from clinical literature and iteratively updated based on newly retrieved evidence.

## C.3 DYNAMIC TOOL SELECTION STRATEGY

Our agent dynamically switches between general and medical-specific tools to ensure complete evidence chains. The tool selection is governed by a learned policy that evaluates query complexity:

$$P(t \mid s_t, q) = \begin{cases} \sigma(\mathbf{w}_m^T \phi(s_t, q)) & \text{if } t \in \mathcal{T}_{\text{medical}} \\ \sigma(\mathbf{w}_g^T \phi(s_t, q)) & \text{if } t \in \mathcal{T}_{\text{general}} \end{cases}$$

where $\phi(s_t, q)$ extracts features that include the rarity of the entity, the required reasoning hops, and the presence of medical terminology, $\mathbf{w}_m$ and $\mathbf{w}_g$ are learned weight vectors, and $\sigma(\cdot)$ is the sigmoid function. The policy learns to prioritize medical tools when encountering rare diseases or complex chemical compounds while leveraging general tools for contextual information.

# D TRAINING IMPLEMENTATION DETAILS

## D.1 SUPERVISED FINE-TUNING CONFIGURATION

**Dataset.** We train on $\mathcal{D} = \{(x^{(i)}, y^{(i)})\}_{i=1}^{N}$ where $N = 2,137$ trajectories, with $x^{(i)}$ denoting input context and $y^{(i)}$ the expert action sequence. The objective maximizes:

$$\mathcal{L}_{\text{SFT}}(\theta) = -\frac{1}{N} \sum_{i=1}^{N} \sum_{k=1}^{|y^{(i)}|} \log p_\theta(y_k^{(i)} \mid x^{(i)}, y_{<k}^{(i)}) \tag{1}$$

**Robustness Augmentations.**

- **Tool failure simulation:** 5% random corruption of tool outputs to encourage error recovery
- **Intermediate thought supervision:** Explicit reasoning traces before each tool invocation
- **Multi-task sampling:** Balanced batching across diagnosis (30%), treatment (25%), guidelines (25%), rare diseases (20%)

**Optimization.**

- Optimizer: AdamW with $\beta_1 = 0.9$, $\beta_2 = 0.98$
- Learning rate: $\lambda = 0.01$ with cosine annealing to $\eta_{\min} = 3 \times 10^{-7}$
- Batch size: 128 (16 per GPU $\times$ 8 H800 GPUs)
- Training epochs: 3
- Gradient clipping: 1.0
- Warmup steps: 100

## D.2 REINFORCEMENT LEARNING CONFIGURATION

**Reward Components.** The composite reward function $r = \alpha r_{\text{task}} + \beta r_{\text{expert}} - \gamma r_{\text{efficiency}}$ comprises:

- $r_{\text{task}}$: Binary task completion (1.0 for correct, 0.0 for incorrect)
- $r_{\text{expert}}$: GPT-4 preference score $\in [0, 1]$ evaluating medical accuracy and completeness
- $r_{\text{efficiency}}$: Penalty for redundant tool usage, computed as:

$$r_{\text{efficiency}} = 0.1 \times n_{\text{redundant}} + 0.2 \times n_{\text{post-answer}} + 0.15 \times n_{\text{irrelevant}} \tag{2}$$

**GRPO Configuration.** The GRPO objective:

$$\mathcal{L}_{\text{GRPO}} = \mathbb{E}_{(x,y)\sim\mathcal{D}} \left[ \log \pi_\theta(y|x) \cdot \left( r(x,y) - \bar{r}_{\mathcal{G}(x)} \right) \right] \tag{3}$$

where $\bar{r}_{\mathcal{G}(x)}$ is the group-level baseline (batch average).

- Group size: 4 responses per query
- Sampling temperature: 0.7
- PPO clip range: 0.2
- Value loss coefficient: 0.5
- Entropy coefficient: 0.01
- Training iterations: 500
- KL regularization: Disabled (following He et al. (2025))

**Curriculum Learning.** Task complexity increases based on moving average pass rate:

- Level 1 (pass rate > 80%): Single-hop queries
- Level 2 (pass rate > 60%): 2-3 hop queries
- Level 3 (pass rate > 40%): 4+ hop queries with rare entities

# E ABLATION STUDY DETAILS

Table 3: Ablation study for MedResearcher-R1. We remove key components while keeping all other settings fixed. Statistical significance: $^*$ $p<0.05$ vs. the Full model. MedBrowseComp is reported as # correct out of 50.

| Model Configuration | MedBrowseComp (correct / 50) | GAIA (%) | XBench (%) | Avg. Tool Calls |
|---|---|---|---|---|
| **MedResearcher-R1 (Full)** | **27.5** | **53.4** | **54.0** | 4.2 |
| *Component Ablations* | | | | |
| w/o Medical Tools | 23.1 | 48.3 | 40.0 | 3.3 |
| w/o RL Training (SFT only) | 25.5 | 50.2 | 51.0 | 3.7 |
| w/o MTG | 24.2$^*$ | 44.3$^*$ | 47.8$^*$ | 3.5 |
| w/o Rare Entities | 20.1$^*$ | 27.8$^*$ | 38.2$^*$ | 3.2 |
| *Data Ablations* | | | | |
| Common Entities Only | 23.0$^*$ | 43.0$^*$ | 46.0$^*$ | 4.5 |
| No Tool Diversity | 21.0$^*$ | 38.0$^*$ | 49.0 | 3.2 |
| *Training Ablations* | | | | |
| SFT Only | 25.5$^*$ | 49.0 | 48.0$^*$ | 3.4 |
| RL Only (no SFT) | 12.0$^*$ | 34.0$^*$ | 34.0$^*$ | 3.2 |

**Tasks and metrics.**

- **MedBrowseComp** is reported as *correct/50*.
- **GAIA** and **XBench-DeepSearch** follow official (%) scoring.
- **Avg. Tool Calls** is the average number of tool invocations per example.

**Evaluation protocol.** All ablations share the same backbone, prompts, decoding parameters, tool budgets, and evaluation splits as the Full model; only the targeted component is removed/altered. Each number is the mean of three seeds.

**Significance.** We compute paired bootstrap (over instances) against the Full model with 10,000 replicates; we mark $^*$ for $p<0.05$.

