# OpenReview forum: "MedResearcher-R1: Expert-Level Medical Deep Researcher via A Knowledge-Informed Trajectory Synthesis Framework"
_ICLR.cc/2026/Conference — ICLR 2026 Conference Withdrawn Submission_

### Official Review · Reviewer_hBVx · 2025-10-23

**Soundness:** 2
**Presentation:** 2
**Contribution:** 3
**Rating:** 4
**Confidence:** 4

**Summary:**

This paper presents MedResearcher-R1, a domain-specialized deep research agent for complex medical question answering and evidence synthesis. Central to its framework are (1) the Knowledge-Informed Trajectory Synthesis Approach (KISA), which constructs knowledge graphs from rare medical entities to generate challenging multi-hop reasoning examples, and (2) a proprietary medical retrieval engine integrated into a dynamic tool-use architecture. Its training stages include Masked Trajectory Guidance and RL optimization and the model is evaluated across medical and general deep research tasks, claiming state-of-the-art performance, notably surpassing leading proprietary and open-source agents on MedBrowseComp.

**Strengths:**

1. The KISA framework for trajectory and dataset synthesis brings ambitious structure and challenge to the often-sparse training of medical research agents. Leveraging rare medical entities and graph-based reasoning, the framework produces questions that simulate authentic, expert-level medical inquiries rather than trivial lookups.
2. The introduction of a private medical retriever elevates retrieval fidelity, while the dynamic tool selection policy usefully blends domain-specific and general resources.
3. Multiple benchmarks (MedBrowseComp, GAIA, XBench) are used, with MedResearcher-R1 outperforming well-known baselines. The ablation study is well-structured, demonstrating the additive impact of rare-entity inclusion, RL, the medical tools, and masked guidance.

**Weaknesses:**

1. The lack of direct baseline comparisons with professional biomedical large models weakens the analysis. In the manuscript, classifying "Gemini-2.5-Pro-deepsearch" and "o3-deepresearch" as Medical Domain Baselines is inaccurate; genuine Medical Domain Baselines should be incorporated.
2. The main body of the paper presents a deep research model in the medical domain; however, its evaluation in the medical field is limited to the MedBrowseComp benchmark, raising concerns about its generalizability and robustness.
3. It is asserted that medical training with rare, complex queries imparts transferable general problem-solving benefit, supported by good but not state-leading scores in open-domain tasks (GAIA, XBench). Yet, Table 2 demonstrates that, despite being competitive with WebSailor-32B, MedResearcher-R1 is outperformed by WebSailor-72B and does not claim top scores. Claiming that "medical training enhances versatility" may be overstated based on the present results, as the improvements observed may stem from backbone size, training regimen, or other ancillary factors.
4. There is a notable absence of detailed error breakdowns (e.g., types of errors made by MedResearcher-R1 versus general agents on MedBrowseComp). The system's failure cases are not systematically studied. Do failures cluster around certain specialties, path complexities, or tool types?

**Questions:**

1. Could authors provide a comparison and analysis of the performance of MedResearcher-R1 and other baselines on tasks involving rare diseases diagnosis, such as RareBench (X Chen et al., 2024), as well as expert-level medical challenges, such as the Medicine domain in Humanity's Last Exam (L Phan et al., 2025)?
2. In the process of constructing complex problem trajectories, KISA employs OpenAI-o3 deep research and GPT-4 to regulate their accuracy at approximately 50%, thereby ensuring the difficulty and complexity of the problems. However, this raises the question: how is the correctness of the problems and trajectories guaranteed?
3. Please clarify the "No Tool Diversity" and "Common Entities Only" ablation settings in Table 3. What was controlled/replaced in these ablation setups to systematically test the effect?

---

### Official Review · Reviewer_Kcrw · 2025-10-25

**Soundness:** 3
**Presentation:** 3
**Contribution:** 3
**Rating:** 6
**Confidence:** 3

**Summary:**

This paper introduced a medical deep research model. The authors first construct knowledge graphs around mined rare medical entities and extract long reasoning chains as complex multi-hop question-answer pairs. Then the authors construct a private medical retrieval to access authoritative medical databases, alongside general-purpose tools. Based on the generated diverse trajectories, the authors train a MedReasearcher-R1-32B via supervised fine-tuning and GRPO with composite rewards. The model achieved strong results on a medical deep research benchmark and two general-purpose multi-domain benchmarks.

**Strengths:**

S1: Medical deep research is an important question to study and requires specialized training and tool efforts.
S2: The paper made several contributions, including KISA, the trajectory synthesis approach, and MedResearcher-R1, through large-scale agent training.
S3: The evaluation results look strong.

**Weaknesses:**

W1: Although the authors promised to release code and models in the reproducibility statement, there is no anonymous link or code uploaded in the supplementary material. It would be very valuable to see them open-sourced.
W2: The related work lacks enough coverage of the ai agents in the medical domain, such as MedAgentGym and MedAgentBench. Please also consider testing MedResearcher-R1 on those benchmarks.

**Questions:**

Q1: How exactly are the probabilities computed in the Clinical Reasoning Engine in Appendix C.2 using Bayesian inference? i.e., how to calculate P(s_i | D_j, c) and P(D_j | c)
Q2: Regarding the neighbor expansion in section 3.1.1, are the neighbors also rare medical entities? And how exactly does Discover() work by searching for novel entities? Are these novel entities also rare?
Q3: Are all the entities in the reasoning graph path masked? If yes, is it still possible for the trained model to navigate according to the masked input path?
Q4: How exactly does the hybrid data strategy in Section 3.2.2 create the training set? Where do D_{guided} and D_{exploration} come from?

---

### Official Review · Reviewer_oeVv · 2025-10-30

**Soundness:** 3
**Presentation:** 2
**Contribution:** 2
**Rating:** 2
**Confidence:** 3

**Summary:**

Authors propose a medical search agent called MedResearcher-R1, which is designed to overcome lack of dense medical domain knowledge and insufficient retrieval from authoritative medical sources. Proposed framework includes three key components: (i) Knowledge-Informed Trajectory Synthesis (KISA) which is used to construct complex multi-hop reasoning trajectories around rare medical entities, (ii) a private medical retrieval engine (FDA, PubMed, trial registries, etc.) with custom tools, and (iii) a Masked Trajectory Guidance (MTG) on reasoning trajectories to prevent memorization.
Authors train Qwen-32B with SFT followed by RL (GRPO) for MedResearcher-R1-32B, which achieves 27.5/50 on MedBrowseComp, outperforming other closed source search engines, and maintains competitive general-domain performance on GAIA and XBench.

**Strengths:**

- Authors identifies specific limitations in existing general search engines (lack of dense medical knowledge and inadequate specialized retrieval) and clearly motivate for the problem, which makes the research focus important.
- The core contribution of authors lies in their data generation pipeline which integrates KISA and MTG; where KISA constructs structured knowledge graphs to generate complex multi-hop medical questions with adaptive difficulty calibration and MTG tries to improve reasoning generalization by discouraging shortcut memorization. Such a data generation pipeline can be helpful for training domain-specific LLMs, also beyond the medical scope.

**Weaknesses:**

- The overall framework follows a well explored ReAct + GRPO fine-tuning in agents, where the contribution mainly lies in dataset construction rather than algorithmic innovation.
- The main technical limitation of the paper lies in the lack of detailed experimental and ablation analyses. Although authors conduct their trainings with SFT followed by RLVR, the training dynamics are overlooked; particularly the individual contributions of SFT and the impact of specific reward design choices are not reported. Especially, the absence of reward analysis, which is the most critical aspect of RLVR, makes it extremely difficult for reviewers to fully understand the inner workings and improvements of the proposed framework.
- The main results (Table 1 and Table 2) lack comprehensive comparisons; for example, it remains unclear how Search-R1 or the base model Qwen-32B perform on MedBrowseComp, which would contextualize the reported gains. Similarly, in Table 2, only the proposed MedResearcher-R1 is evaluated as a search model on XBench-DeepSearch and GAIA, while performances of other models listed in Table 1 (e.g., o3-deepresearch, Gemini-2.5-Pro) are not reported. These omissions make it difficult to understand the effectiveness of MedResearcher-R1 and fairly compare it with other baselines.

Overall, eventhough data-generation part is promising, the absence of more technical discussions and deeper analysis makes the current version of the paper seems more like a technical report.

**Questions:**

1. What is the contribution of each reward component, and how does the model’s medical reasoning evolve throughout training?
2. How does cold-start training affect convergence or stability?
3. Can the authors provide qualitative comparisons; for instance, contrasting reasoning rollouts and final answer responses between MedResearcher-R1, Search-R1, and o3-deepresearch to better illustrate behavioral differences and reasoning quality on challenging medical scenarios?

---

### Official Review · Reviewer_AGH9 · 2025-11-06

**Soundness:** 3
**Presentation:** 2
**Contribution:** 2
**Rating:** 4
**Confidence:** 5

**Summary:**

MedResearcher-R1 is an agent that can do complex clinical reasoning by integrating with reliable knowledge sources. For doing this, the authors first developed a novel KISA approach to generate complex questions by building knowledge graph based on entities occurring in PubMeds. And then fine-tune the LLM with SFT and RL to achieve the comparable performances with much larger LLMs, but only with a limited training dataset.

**Strengths:**

1. The topic is very interesting and the way to create the knowledge graph is also interesting
2. I believe that think-verify is a good way to get better answer and also SFT is needed in medical reasoning.
3. The model outperforms all baselines only with a small training dataset, which is really exciting.

**Weaknesses:**

1. All figures are confusing.
2. It is a little bit hard to follow.
3. Some parts need further clarification.
4. Please clarify on training dataset and the objects in each training stage.

**Questions:**

1. Could you please provide more details on how to avoid typos when you extract the entities from PubMed abstracts?
2. How did you create the knowledge graph? Identify the medical terminologies from abstract and their relationship as well and then create the knowledge graph. I just have a question, how did you solve the drugs with different brand name? or the same entities with different presentations?
3. The authors claimed that the generated questions require synthesis across multiple medical information sources. However, it seems you only  adopt PubMed as the knowledge source. Why did the authors claim it as multiple sources? In addition, I am wondering how do you solve knowledge conflicts during the construction?
4. In terms of the MTG, I am curious about two key behaviors the authors mentioned in the paper. From my understanding and the figure 1, could you please provide us an example what is the meaning of when to use tools? In addition, your framework is first to generate and then verify, right? So what does the mean of when to use tools? I don't think you are optimizing when to use tools, but just decide which tool to call.
5. In the training dataset for MTG, did you mask all head and tail entities in a reasoning path? or just randomly mask some entities? According to the formulation {([MASK], r_i, [MASK])} from i to n-1, it seems you masked all entities, so what you wish is to use tools to recover these triplets?
6. The exploration dataset: 1) could you please provide a scenario that needs to switch between general and medical tools? 2) the error, does this mean that you introduce errors and then to test if your model could correct this error? And in both guided and exploration dataset, each of them is a set of makers reasoning path? or real reasoning path?
7. Figure 4 doesn't make sense. First of all, this is not a complete reasoning trace. Second, the authors use the verify word for the first time throughout this paper. I cannot know what you are verifying. If all the output is based on tools, is it necessary to verify? In addition, your training dataset is composed by a sequence of triplets, why here you are discussing the paradigm of iterative search-verification-synthesize paradigm. I am so confusing here. I think consistency checking is better than verification in Figure 4, because you just want to confirm if the knowledge from multiple sources are consistent.
8. I am quite confused on the MTG. From my understanding, MTG is to mask entities. so the training dataset is a sequence of triplets and each triplet is ([MASK], r, [MASK]). And this dataset is used in SFT. What is the model learning? if all triplets are [MASK]. I think the authors should make it clear. First, please provide an example in training dataset, and then please tell us the D_guided is composed by graph path or masked graph path. What is the input data format in SFT stage?
9. Please clarify the optimized object in each stage. In SFT stage, if the trajectory contains the tool call? what is the expert action sequence? is it a sequence of triplets or the ground truth answer? In RL stage, what is the objective function? to make the LLM can generate the correct answer? since all rewards are depended on answer no reasoning trace.

---

### Note · Authors · 2025-12-30

I have read and agree with the venue's withdrawal policy on behalf of myself and my co-authors.